# Prevalence of Static Balance Impairment and Associated Factors of University Student Smartphone Users with Subclinical Neck Pain: Cross-Sectional Study

**DOI:** 10.3390/ijerph191710723

**Published:** 2022-08-28

**Authors:** Saw Wah Wah, Uraiwan Chatchawan, Thiwaphon Chatprem, Rungthip Puntumetakul

**Affiliations:** 1Human Movement Sciences, School of Physical Therapy, Faculty of Associated Medical Sciences, Khon Kaen University, Khon Kaen 40002, Thailand; 2Research Center of Back, Neck, Other Joint Pain and Human Performance (BNOJPH), Khon Kaen University, Khon Kaen 40002, Thailand; 3School of Physical Therapy, Faculty of Associated Medical Sciences, Khon Kaen University, Khon Kaen 40002, Thailand

**Keywords:** balance error scoring system, smartphone users, daily hours of smartphone use, years of smartphone use and neck disability index score

## Abstract

The aim of this study was to assess the prevalence of static balance impairment in university student smartphone users with subclinical neck pain and identify the associated risk factors. Because of rapid and widespread smartphones use, and the subsequent effect on neck pain in university students, it is essential to determine the prevalence of balance impairment and associated factors in this population. Simple random sampling was completed among eighty-one participants in this cross-sectional study. A self-reported questionnaire, fitted precisely for smartphone users, was used prior to clinical assessment by the Balance Error Scoring System. Both simple and multiple logistic regressions were used to analyze the prevalence of static balance impairment and associated factors. The prevalence of static balance impairment in university student smartphone users with subclinical neck pain was 74.07% (95% CI: 64.32 to 83.82). The significant risk factors were “daily smartphone use ≥ 4 h’’ (AOR: 19.24 (95% CI 4.72 to 78.48) *p* = 0.000), “≥4 years of smartphone use” (AOR: 5.01 (95% CI 1.12 to 22.38) *p* = 0.035), and “≥7 neck disability index score’’ (AOR: 12.91 (95% CI 2.24 to 74.45) *p* = 0.004). There was a high prevalence of static balance impairment in university smartphone users with subclinical neck pain. University student smartphone users with subclinical neck pain who met at least one of the risk factors should realize their static balance impairment.

## 1. Introduction

In modern society, multifunctional smartphones are frequently used in daily life. Smartphones serve not only as a multimedia collection, camera lens, and global and satellite navigation system [1] but also as a means for sending and accepting email, storing data, playing games, and engaging in learning interactions [2]. These activities have led to a rapid increase in worldwide smartphone use and concomitant neck pain [3] and have altered users’ dynamic balance [4,5,6].

Among smartphone users, university students in their 20s use their smartphones more than any other age group [7]. In the United States, in 2017, approximately 96% of people aged 18 to 24 used mobile phones 2.5 h per day, and 97.7% of the mobile phones used were smartphones [8]. In Hong Kong and Thailand, smartphone use triggered neck pain in 68.2% and 90%, respectively, of 18 to 24-year-old university student smartphone users [9]. This neck pain incidence was attributed to awkward postures and static and repetitive work [10].

Subclinical neck pain (SNP) refers to mild-to-moderate recurrent neck pain. Sub-clinical means that the participants have not yet taken treatment as the pain severity is not severe. Stages of SNP may be within 7 days to 3 months (subacute SNP) or within more than 3 months (chronic SNP) [11,12,13]. It is classified according to the area identified by a subject who complains of pain on both sides of the neck, including central of the posterior part of the cervical spine from the superior nuchal line to the T1 spinous process, as central SNP. Right SNP is identified by a subject who complains of neck pain only on the right side of the neck and left SNP only on the left side of the neck [12].

Smartphone users with neck pain flex their neck slightly more than those without neck pain while using their smartphone [14]. Almost 91% of college students position their neck in flexion throughout smartphone use [15]. As neck flexion increases, there is increased compressive load on neck structures [16]. The severity of musculoskeletal symptoms is also associated with daily hours of smartphone use [17]. A total time of smartphone use ≥ 2 h per day is associated with a neck and shoulder pain adjusted odds ratio (AOR) = 1.49 (95% confidence interval (CI) = 1.20–1.86) [18], and total time spent on smartphone ≥ 2.4 h per day is associated with a neck pain OR = 2.27 (95% CI = 1.24–5.96) [19]. Previous studies indicated a moderate-to-strong association between the total time spent on smartphones and neck pain with an AOR ranging from 1.49–8.63; variations occurred according to the total time spent on smartphone, postures, and other factors [20]. Total time spent, postures, and other factors decrease cervical proprioception and dynamic balance control in smartphone users [6]. 

Dual tasks using smartphones [4], short note sending [5], and prolonged use > 4 h/day negatively affect dynamic balance control [6]. The maintenance of balance is essential in the prevention of injuries, and it depends on the central nervous system through the integration of sensory information from the vestibular, somatosensory, and visual systems [21]. Alterations in any of these inputs disturb balance and increase the risk of injury [22]. Although smartphone users encounter problems associated with dynamic balance control [4,5,6], most of a person’s daily activities that commonly include static balance control consume minimal muscle energy [23]. Static balance is the capability of orientating the center of mass over the base of support while the body is at rest [24]. The correlation between anthropometrical characteristics and static and dynamic balance is proven in the study of Tabrizi, H.B. et al. (2013) [25].

Long-duration smartphone use increases cervical joint position sense error [6,7,26], visual fatigue [27,28], neck pain [20], and adopted flexed neck posture [29,30,31], while decreasing dynamic balance [4,5,6]. Alterations in visual, sensory, and proprioceptive input to the central nervous system might be related to static balance in smartphone users. Previous studies have not yet determined the prevalence of static balance impairment in smartphone users. There is a lack of knowledge on the effects of smartphone use on static balance and factors associated with balance impairment in relation to smartphone use.

The current study was aimed at evaluating the prevalence of the static balance impairment of university student smartphone users with SNP and factors associated with static balance impairment in this population. It was expected that there might be a high prevalence of balance impairment in smartphone users and that daily hours of smartphone use might be high and impact static balance control. The findings from this study may prove useful in preventing impaired static balance control and recurrent neck pain in smartphone users.

## 2. Materials and Methods

### 2.1. Design

Simple random sampling was conducted in this cross-sectional study at The Laboratory of Physiotherapy, Department of Physiotherapy, University of Medical Technology, Yangon, Myanmar. This study was conducted in accordance with the Declaration of Helsinki and was approved by the Khon Kaen University Ethics Committee for Human Research (HE 612374) and Thai Clinical Trial Registry (TCTR20190903003).

### 2.2. Participants

University student smartphone users with neck pain at Yangon University of Medical Technology, Myanmar were invited to participate in this study. The inclusion criteria are listed as follows: (1) subclinical or intermittent neck pain and mild-to-moderate neck dysfunction without regular treatment; (2) 18–25 years of age; (3) BMI ≤ 30 kg/m^2^, as BMI > 30 kg/m^2^ could negatively influence balance control [32]; (4) experience in using a smartphone with > 6 months duration and daily smartphone screen time ≥ 2 h [19,33]; (5) neck disability index (NDI) score range of 5–14/50; (6) mild to moderate pain on the visual analogue scale (VAS) 30 to 74 mm; and (7) voluntary participants who comprehend English.

Participants with any one of the following conditions were excluded: (1) visual, auditory, vestibular or neurological deficits; (2) traumatic injuries or surgical interventions of the spine and lower limb within one year prior to the beginning of the study; (3) medical conditions which may have a negative effect on balance; (4) chronic musculoskeletal diseases, lower limb fractures, and injuries; (5) participation in any neck muscle strengthening and balance training over the past 12 months; (6) Beck Depression Inventory (BDI) score > 30/63; (7) Dizziness Handicap Inventory (DHI) score > 30/100; and (8) sedative drug or alcohol use within the past 48 h.

### 2.3. Screening and Experimental Process

Participants were recruited via poster invitation on notice boards of each Faculty at University of Medical Technology, Yangon. A physiatrist professor and the researcher (Saw Wah Wah, SWW) assessed each participant based on a specification for signs and symptoms of SNP [34], a self-reported questionnaire for smartphone users, and criteria for eligibility. All participants provided written informed consent. For optimum measurement quality, prior to the test trials, familiarization of eligible participants was conducted. This process involved 2 sets of 10 s each for the Balance Error Scoring System (BESS) elements, 2 sets of 10 s each for the craniocervical flexion test (CCFT), 2 sets of 5 s each on each cervical joint position sense (CJPS), and 2 sets of 1 s each for the VAS. Participants were then assessed for formal BESS, CJPS, CCFT, VAS, and NDI measurement. Figure 1 shows a flow diagram of the participants in this study.

### 2.4. Outcome Measurements

#### 2.4.1. Static Balance 

The Balance Error Scoring System (BESS) was the principal outcome measure to study static balance. The BESS is a continuous score measure; lesser BESS scores represent superior balance control, whereas greater scores represent inferior balance control on a total 0–60 BESS scores. For participants aged 18–25, the total BESS score of ≥ 15 is the cutoff point for balance impairment [33,35]. The BESS is a standardized clinical measure to assess balance impairment [33,36]. The BESS has exhibited adequate to excellent validity with force-plate target sway (r = 0.31 to 0.79) and moderate reliability (intraclass correlation coefficient, ICC = 0.07) to excellent (ICC = 0.92) reliability [33,37].

To assess static balance, participants were instructed to maintain standing balance barefoot with hands on their hips while performing each of the six BESS sub-tests with closed eyes for 20 s. The sub-tests of the BESS are described as follows: (a) Double leg stance with feet together (firm surface), (b) single leg stance on non-dominant foot (firm surface), (c) tandem stance on non-dominant foot in back (firm surface), (d) double leg stance with feet together (foam surface), (e) single leg stance on non-dominant foot (foam surface), and (f) tandem stance on non-dominant foot in back (foam surface) (Figure 2) [33]. Each sub-test was recorded by counting participant’s errors of deviance from the selected test position. In case of several concurrent errors, the failure to maintain the selected test position for >5 s was counted as an error. Errors included hands-off iliac crests, eyes opening, fall or footstep stumble, abduction or flexion of hip > 30 degrees, and lifted forefoot or heel from test surface. The total BESS score was calculated by adding the number of errors per each of the six sub-tests [35,38]. We controlled the hip angle of the raised (dominant) side in the SLS sub-test by thoroughly instructing the participant to maintain their hip angle in neutral with the flexed knee at 90 degrees [33]. 

In recent research, SWW was trained by an expert physiotherapist, Dr. Rungthip Puntumetakul (Dr. RP) from Khon Kaen University, Thailand, with 30 years of clinical experience. SWW was trained how to use the CROM, including the protocol, score card, normal value, cutoff scores, and confounding factors influencing measurement. Training proceeded until the expert physiotherapist was satisfied with the accuracy of the results from the researcher. This study showed excellent intra-rater reliability (ICC = 0.98 to 0.99).

#### 2.4.2. Cervical Joint Position Sense

CJPS, or cervical proprioception, was measured with a cervical range of motion measurement (CROM) device, Deluxe USA, EN-121156 [39]. CROM has been indicated to have excellent concurrent validity (r = 0.93–0.98), excellent inter-rater reliability (ICC = 0.89–0.98) with 3D Fastrack [40] and excellent correlation (r = 0.78–0.86), and significant-to-excellent test–retest reliability (ICC = 0.74–0.96) with the VICON motion capture system [41]. CJPS error ≥4.5 degrees shows cervical proprioception impairment [42].

To measure cervical proprioception, a CROM device was put on the participant’s nose-bridge and ears and a Velcro strap was secured to the head. The natural head posture (NHP) was selected to indicate zero degrees at each of the horizontal, sagittal, and compass meters. Each participant was blindfolded and sat upright on a chair, with hands on thighs, feet on floor, flexed hips, and knees at 90 degrees. Rehearsal to recognize the CJPS, NHP, and relocation to NHP was accomplished for 5 s prior to the assessment trials [43,44]. Three trials of relocation to NHP from each cervical range of motion was averaged and then noted in degrees as CJPS [39].

SWW was trained by Dr. RP about how to conduct the CROM device, including the protocol, how to position the subject in the NHP, how to reinforce the accuracy by 2 magnets placed over the subject’s shoulders, and how to identify and calculate the CJPS degrees, score card, normal value, cutoff scores, and confounding factors influencing measurement. Training advanced until the expert physiotherapist was satisfied with the accuracy of the results from the researcher. Intra-examiner reliability of this study ranged from ICC 0.78 to 0.99.

#### 2.4.3. Craniocervical Flexors Function Test

Muscle function, strength or muscle endurance of the deep cervical muscles was assessed with the craniocervical flexors function test (CCFT) [45]. The highest pressure from a baseline of 20 mmHg at which the participant could accurately perform the CCFT up to 10 s was the activation pressure score. The highest target pressure that the participant could attain and hold for 10 s, starting from 20 mmHg with an increase of 2 mmHg at each phase, with a total possible increase in 5 phases up to 30 mmHg (target pressures of 22, 24, 26, 28, and 30 mmHg), was the performance pressure score. If a participant could complete the third level of the test (at 26 mmHg) and achieve 7 repetitions of correctly holding the CCFT for 10 s, then the performance pressure score was 6 × 7 = 42 mmHg. Probable performance pressure scores of the CCFT ranged from 0 to 100 mmHg [45,46,47]. Dysfunction of the craniocervical flexors was indicated by a decreased performance pressure score of CCF ≤ 24 mmHg [48]. 

The participant was assessed by the CCFT in the crook lying position. A stabilizer or feed-back device (Chattanooga Group, Inc., Hixson, TN, USA) with a towel were positioned under the participant’s suboccipital area. The physiotherapist instructed the participant in the CCFT as cervical flexion of the upper cervical spine without any further flexion of the middle or lower cervical spine. Two types of CCFT scores were measured in two tests: activation pressure score and performance pressure score [45].

In the present research, SWW was assessed by an expert physiotherapist, Dr. RP, to confirm that she was qualified to competently use the feed-back device “stabilizer”, including knowledge of the protocol, positioning the subject, identifying normal versus abnormal activity of the craniocervical flexors, use of the score card, normal value, and cutoff scores and avoiding confounding factors influencing measurement. Training stopped when the expert physiotherapist was satisfied with the accuracy of the results from the researcher. The intra-rater reliability of this study (ICC) ranged from 0.91 to 0.99.

#### 2.4.4. Visual Analogue Scale 

A VAS was used to report the participant’s intensity of pain on a 100 millimeters (mm) scale. The VAS is an easy patient-reported measure [49]. A greater VAS score shows larger intensity of pain. The cutoff points were no pain (0–4 mm), mild pain (5–44 mm), moderate pain (45–74 mm), and severe pain (75–100 mm) [50,51].

In the VAS assessment, the participant placed a vertical line on the visual analogue scale line at the point that reflected his/her pain intensity. The distance in millimeters from the zero anchor was documented as the VAS of the participant [49,52].

#### 2.4.5. Neck Disability Inventory Score

The NDI is a patient-reported measure to assess neck pain due to daily life and disability [53]. The NDI is a specific conditional and functional questionnaire including 10 items: pain, personal care, lifting, reading, headaches, concentration, work, driving, sleeping, and recreation [54]. The NDI shows excellent validity with other instruments [55,56] and excellent intra-rater reliability (ICC = 0.93) in patients with neck pain [57].

Each of the 10 items is scored from 0–5, in which zero means “no pain” and 5 means “worst imaginable pain”. The maximum score is 50. The duration of the test is from 3 to 7.8 min [54].

The cut-off scores can be interpreted as follows:

0–4 points: no disability;5–14 points: mild disability;15–24 points: moderate disability;25–34 points: severe disability;35–50 points: complete disability.

A higher score indicates greater patient-rated disability. The minimal clinically important change (MCID/MCIC) was calculated as 5 points [55].

#### 2.4.6. Dizziness Handicap Inventory

The DHI was developed to measure the self-perceived level of handicap associated with the symptom of dizziness [58]. The DHI has 25 items with 3 response levels, sub-grouped into three domains: functional, emotional, and physical. The DHI has good correlation with specific objective measures of balance [59,60,61]. The DHI may be useful in identifying subjects with benign paroxysmal positional vertigo [61]; a total score of 0 to 30 indicates a mild dizziness handicap, a total of score of 31 to 60 indicates a moderate dizziness handicap, and a total score of 61 to 100 indicates a severe dizziness handicap [61]. The DHI is reliable (r = 0.92 to 0.97) and is a valid, comprehensively, and clinically useful tool to measure self-perceived handicap associated with the symptom of dizziness from a variety of causes [62]. The participants were screened with DHI questionnaires to exclude vestibular disorder if their DHI score > 30/100.

#### 2.4.7. Beck Depression Inventory 

The BDI is a 21 item, self-report rating inventory that assesses characteristic attitudes and depression symptoms; each item corresponds to a major depressive symptom in the preceding 2 weeks [63]. Internal consistency for the BDI ranges from 0.73 to 0.92 with a mean of 0.86 [64,65]. The BDI demonstrates high internal consistency, with alpha coefficients of 0.86 and 0.81 for psychiatric populations and non-psychiatric populations, respectively [64]. The BDI lasts approximately 10 min to understand the questions comprehensively and sufficiently [65]. In the present study, the BDI was used as a screening device if a participant’s BDI score > 30/63 to exclude depression symptoms. 

### 2.5. Sample Size

The sample size estimation was conducted after the pilot study. When *p* (prevalence) was greater than or equal to 0.6 and 95%CI (1.96), d (precision) could be set to 0.1 [66]. The sample size of the study was 81. It was considered acceptable as described by Naing et al. (2006) [67] and Puntumetakul et al. (2022) [68]. 

### 2.6. Statistical Analysis

Participant characteristics and variables were analyzed using descriptive statistics. Continuous variables such as age, body mass index (BMI), and BDI scores were analyzed by the mean and standard deviation (SD). Categorical variables were analyzed using frequency and percentage. Simple logistic regression analysis was employed to calculate the AOR and their 95% CI for the presence of balance impairment for each risk factor. After the simple logistic regression analysis, variables whose *p*-value < 0.20 were taken as candidate variables for the multiple logistic regression. A backward stepwise method for variable selection of multiple logistic regression analysis was used [69]. The *p* values < 0.05 were considered statistically significant. The STATA program version 13.1 (STATA, College Station, TX, USA) was employed in this study.

## 3. Results

### 3.1. Participant Characteristics 

Participant characteristics are described in Table 1.

### 3.2. Impairment: Balance, CJPS and Muscle Function

From a total of 81 smartphone users with neck pain, 60 participants had static balance impairment with a prevalence of 74.07%, (95% CI: 64.32 to 83.82). The prevalence of CJPS impairment (right rotation) was 35.80% (95% CI: 25.14 to 46.47) and that of (left rotation) was 32.10% (95% CI: 21.71 to 42.49). The prevalence of craniocervical flexors function impairment was 72.84% (95% CI: 55.87 to 79.72) (Table 2).

### 3.3. Regression Analysis: Prevalence, AOR, and 95% CI of Balance Impairment with Risk Factors

In a simple logistic regression analysis, fifteen variables reached a *p*-value < 0.2: age 20–25 years; BMI ≥ 20 kg/m^2^; BDI score ≥ 4; DHI score ≥ 7; daily smartphone use ≥ 4 h; ≥4 years of smartphone use; daily ≥ 2 h of other visual display terminal use; moderate neck pain on VAS; NDI score ≥ 7; central pain; chronic pain; ≥ 3 episodes of neck pain; CJPS error on right rotation ≥4.5 degree; CJPS error on left rotation ≥ 4.5 degree; and CCFT ≥ 24 mmHg. These variables were further analyzed by multivariate logistic regression.

In the multiple logistic regression analyses, “daily smartphone use ≥ 4 h’’ (AOR: 19.24 (95% CI 4.72 to 78.48) *p* = 0.000), “≥4 years of smartphone use’’ (AOR: 5.01 (95% CI 1.12 to 22.38) *p* = 0.035), and “NDI ≥ 7 score’’ (AOR: 12.91 (95% CI 2.24 to 74.45) *p* = 0.004) were the risk factors significantly associated with static balance impairment (Table 3).

## 4. Discussion

A new finding from the current study was the high prevalence of static balance impairment (74%) in university student smartphone users with SNP. There are no previous reports for comparison not only regarding prevalence of balance impairment in smartphone users but also on static balance impairment in this population. Possible reasons for the high prevalence of static balance impairment may be associated with the risk factors of smartphone use.

Smartphone use of more than 4 h daily was the most significant risk factor in this study. This knowledge may be useful in preventing impaired static balance control in smartphone users. In both simple and multiple logistic regression analyses, the current study findings demonstrated that a significant risk of static balance impairment occurred when using the smartphone for more than 4 h daily compared with a duration of 2–4 h daily. The current result is consistent with the findings of Azab et al. (2017) [6], which show a significant decrease in dynamic balance among three groups (A, B, and C) of smartphone users with varied hours of smartphone use. The most significant decrease in dynamic balance was found in group C (smartphone use > 4 h per day), which was more than group B (smartphone use <4 h per day) and group A (not smartphone use) [6]. The dynamic balance significantly decreased while dual tasking using a smartphone, such as playing games, sending text messages, web surfing, and listening to music [4]; using a smartphone while standing; and in dual-task situations (sending messages and using a social network service). SNS showed the highest instability. Smartphone SNS involves significant concentration, more than texting or using key-pad mobile phones [5].

Daily mobile phones usage >5 h was significantly associated with cervical and shoulder pain when age and gender were considered. However, a weaker association became obvious after additional confounding factors (age, sex, level of education of parents, school achievement, puberty, and efficacy of physical activity and stress) were adjusted [67]. Previous studies indicated a moderate-to-strong association between total smartphone use hours and cervical pain, with AOR ranging from 1.49–8.63, with variations according to their smartphone use hours, postures, and other factors [17]. Increased total time spent on a smartphone increased neck flexion postures adopted, which occurred in excess of 91% of college student smartphone users [12], resulting in increased visual fatigue [28] and CJPS error [7,18]. The effect of visual input causes changes in muscle activity in the trunk and lower leg muscles during balance control [70]. Thus, the alterations in visual processing can disturb the degree of dependence on proprioception and postural sway or static balance [70]. These issues decreased cervical proprioception and may affect dynamic balance control in smartphone users.

The current results show an association between years of smartphone use and static balance impairment in university student smartphone users. Our study found that more than 4 years of smartphone use had a significant association with impaired static balance, with an AOR of 5.01 (95%CI = 1.12 to 27.38). University students, especially students in their 20s, used their smartphones more than other ages [7]. Myanmar has 53 million citizens, of whom 46 million (85%) are smartphone users [71]. The effects on static balance control by years of smartphone use relate to visual fatigue, which is one component of the balance control system. The long duration of smartphone use induces visual fatigue and negatively effects both static and dynamic balance function. Visual fatigue can be provoked by maintaining 40 cm between the visual display terminal screen and a user’s eyes for an hour [27,28]. Visual fatigue may also substantially affect visual perception feedback and visual focusing ability [72]. Visual recognition is a key factor in maintaining posture and static balance control [73]. Visual input induces changes in muscle activation in the trunk and lower leg muscles throughout balance control [70]. Alterations in visual input can affect the degree of dependence on proprioceptors and postural sway or static balance [70].

Our findings demonstrated that the NDI score was a significant risk factor of static balance impairment in both simple logistic analysis and multiple logistic analysis, with an AOR of 12.91 (95% CI of 2.24 to 74.45). This finding is consistent with those of previous studies with mean NDI scores between 10.6 and 25.8, indicating that mild neck disability was significantly associated with standing balance [74,75,76,77]. Awkward postures, static and repetitive work associated with smartphones use [10], and smartphone addictive scores may be attributed to increased neck disability [78]. When an individual focuses on the smartphone screen, the cervical spine flexes to maintain posture. This shift causes temporary creep in cervical muscles and ligaments from acute neck pain, and inhibitory effects are produced to restrict the use of the affected region [79]. Longer smartphone use results in increased neck pain and NDI scores [80] for university students who use smartphones for periods longer than 3 h daily [81]. Increased adopted flexed neck posture occurred in smartphone users with SNP [26] and in >91% of university student smartphone users [15]. Increased neck flexion angles increase the compressive load on neck structures [16], and pain effects not only nociceptors but also the mechanoreceptors of the spinal cord and the CNS [82,83]. Balance control depends on proprioceptive input from mechanoreceptors and vestibular and visual input to the CNS [84,85,86].

## 5. Limitation

The current study has some limitations. First, because the number of male participants was not equal to the number of female participants, the current study cannot conclude that there was no significant difference in static balance between males and females. Second, the static balance control may be disturbed by physical fitness. Future studies should explore the correlation between static balance impairment and the fitness or activity level. Third, although before conducting the study we calculated the sample size according to prevalence as our main outcome, this sample may not be appropriate when the data were analyzed by multiple logistic regression for determining factors associated with balance impairment as the secondary outcome. Additional research should consider selecting sample size by referencing a multiple logistic regression study to strengthen the results.

## 6. Conclusions

The current study found a high prevalence of static balance impairment in university student smartphone users with SNP. The findings revealed that associations between static balance impairment and ≥4 daily hours of smartphone use, and between ≥4 years of smartphone use and NDI scores ≥7, were statistically significant. Our study found static balance impairment in participants who used their smartphones from 2–4 h daily. Thus, the findings indicate that we should reduce our daily hours of smartphone use, which could negatively impact static balance. Additional research on the arrangements of using smartphones, optimal neck flexion angle, optimal schedule, and management plans to identify the precise mechanisms underlying the associated factors will be beneficial in preventing the adverse effects of smartphone use.

## Figures and Tables

**Figure 1 ijerph-19-10723-f001:**
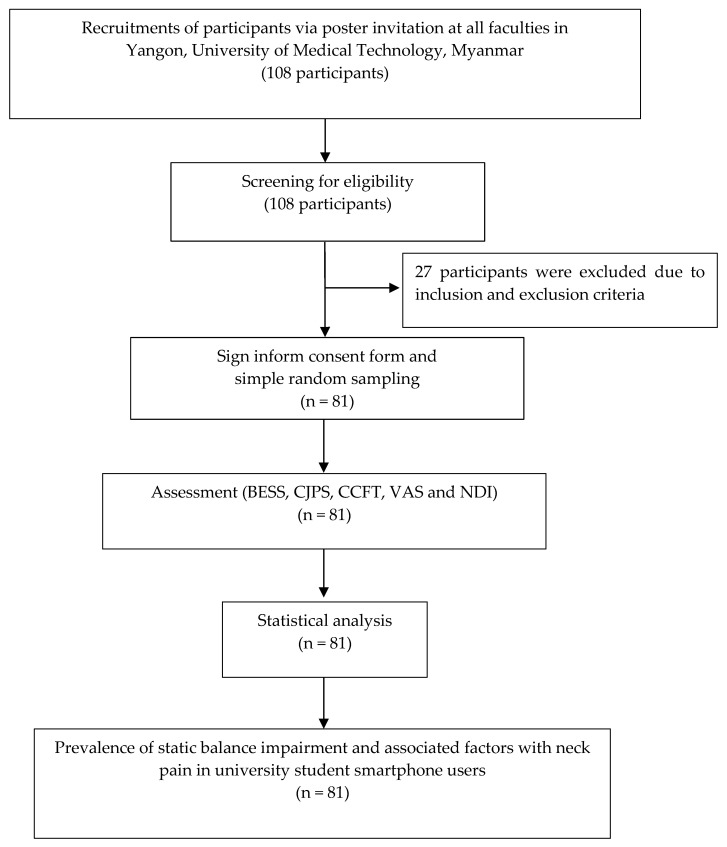
A flow diagram of participation in the study.

**Figure 2 ijerph-19-10723-f002:**
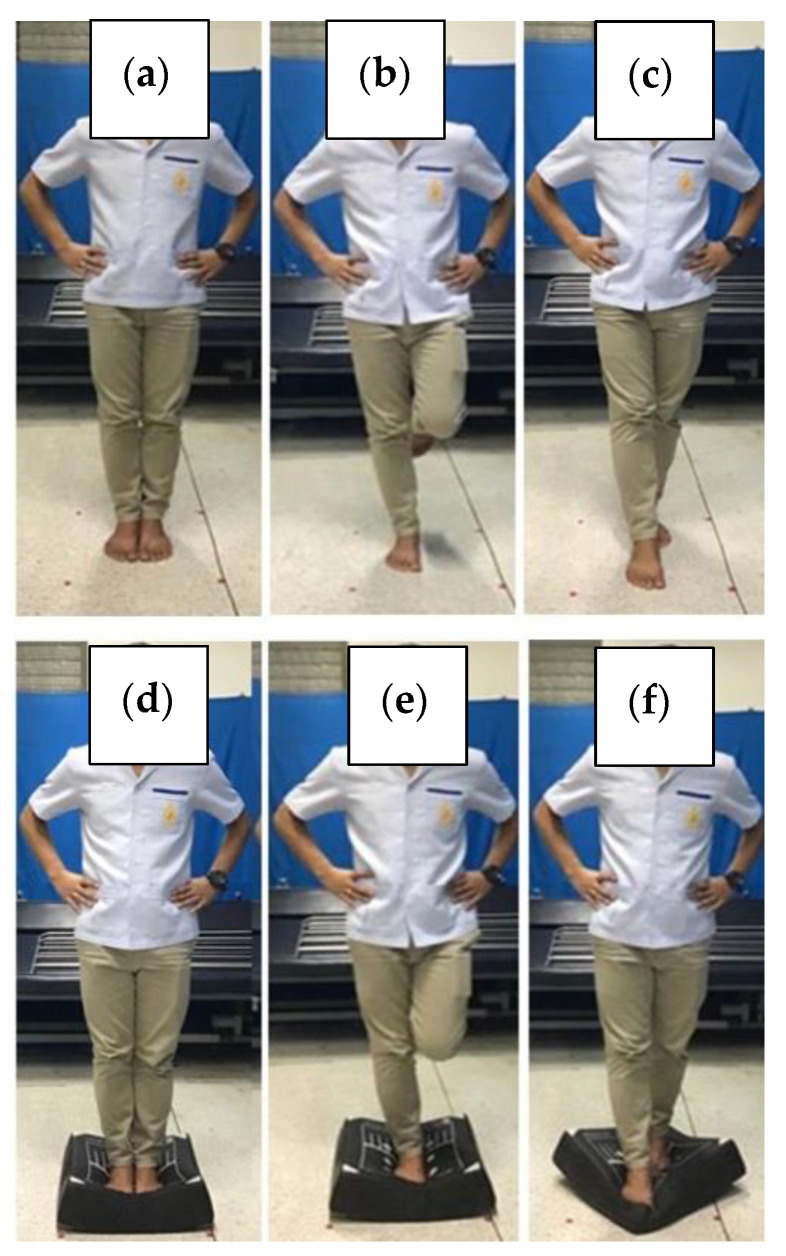
Static balance assessment. (**a**) Double leg stance with feet together (firm surface), (**b**) single leg stance on non-dominant foot (firm surface), (**c**) tandem stance on non-dominant foot in back (firm surface), (**d**) double leg stance with feet together (foam surface), (**e**) single leg stance on non-dominant foot (foam surface), and (**f**) tandem stance on non-dominant foot in back (foam surface).

**Table 1 ijerph-19-10723-t001:** Characteristics of participants (n = 81).

Factors	Total Participants, n (%)	Mean (SD)	Min-Max	Participants without BESS Impairment, n (%)	Participants with BESS Impairment, n (%)
**Demographic**					
Age (years)					
<20 ≥20	45 (55.56) 36 (44.44)	19 (0.51) 21 (1.64)	18–19 20–25	9 (20) 12 (33)	36 (80) 24 (67)
Gender					
Male Female	10 (12.35) 71 (87.65)	-	-	4 (40) 17 (24)	6 (60) 54 (76)
BMI (kg/m^2^)					
<20 ≥20	49 (60.49) 32 (39.51)	18.17 (1.27) 22.45 (2.11)	15.9–19.95 20.3–28.38	10 (20) 11 (34)	39 (80) 21 (66)
BDI (score)					
<4 ≥4	44 (54.32) 37 (45.68)	1.72 (1.93) 8.29 (8.38)	0–8 4–17	14 (32) 7 (19)	30 (68) 30 (81)
DHI (score)					
<7 ≥7	43 (53.09) 38 (46.91)	0.67 (0.07) 0.82 (0.06)	0–6 8–28	14 (33) 7 (18)	29 (67) 31 (82)
**Smart phone and other visual display terminal use**					
Daily hours of smartphone use					
<4 ≥4	24 (29.63) 57 (70.37)	2.88 (0.27) 4.83 (1.70)	2.5–3.5 2.5–8.0	16 (67) 5 (9)	8 (33) 52 (91)
Years of smartphone use					
<4 ≥4	33 (40.74) 48 (59.26)	2.67 (0.48) 4.88 (0.90)	2–3 4–7	15 (45) 6 (13)	18 (55) 42 (88)
Posture when using smartphone					
Sitting Lying	33 (40.74) 48 (59.26)	-	-	11 (33) 10 (21)	22 (67) 38 (79)
Daily hours of other visual display terminal use					
<2 ≥2	60 (74.07) 21 (25.93)	0.38 (0.48) 3.29 (1.82)	0–1 2–9	18 (30) 3 (14)	42 (70) 18 (86)
**Neck pain status**					
Visual analogue scale (mm)					
<44 ≥44	60 (74.07) 21 (25.93)	33.83 (4.37) 50.19 (4.75)	30–44 45–64	21 (35) 0 (0)	39 (65) 21 (100)
NDI (score)					
<7 ≥7	43 (53.09) 38 (46.91)	5.30 (0.46) 9.29 (1.74)	5–6 7–14	19 (44) 2 (5)	24 (56) 36 (95)
Site of neck pain					
Central pain Right or left pain	29 (35.80) 52 (64.20)	-	-	9 (17) 11 (52)	43 (83) 10 (48)
Stage of neck pain					
Subacute Chronic	54 (66.67) 27 (33.33)	-	-	17 (31) 4 (15)	37 (69) 23 (85)
Episode					
<3 ≥3	14 (17.28) 67 (82.72)	-	-	10 (71) 11 (16)	4 (29) 56 (84)
**Cervical joint position sense error** (degrees)					
Right rotation					
<4.5 ≥4.5	52 (64.20) 29 (35.80)	3.05 (0.88) 6.29 (1.84)	1.33–4 3.33–11.3	20 (35) 1 (4)	37 (65) 23 (96)
Left rotation					
<4.5 ≥4.5	55 (67.90) 26 (32.10)	3.03 (0.85) 6.09(1.80)	1.33–4 4.5–11.3	21 (40) 0 (0)	32 (60) 28 (100)
**Craniocervical flexors test** (mmHg)					
<24 ≥24	59 (72.84) 22 (27.16)	26.47 (2.39) 17.64 (3.06)	24–30 12–20	2 (9) 19 (32)	20 (91) 40 (68)

Abbreviations: SD, standard deviation; BMI, body mass index; DHI, Dizziness Handicap Inventory; BDI, Beck Depression Inventory.

**Table 2 ijerph-19-10723-t002:** Prevalence of impairment: balance, cervical joint position sense, and muscle function.

Impairment	Number	Prevalence	95% CI
Static balance impairment	60	74.07	64.32 to 83.82
Cervical joint position sense impairment			
Right rotation	29	35.80	25.14 to 46.47
Left rotation	26	32.10	21.71 to 42.49
Craniocervical flexors function test impairment	59	72.84	55.87 to 79.72

**Table 3 ijerph-19-10723-t003:** Prevalence, adjusted OR and 95% confidence interval of balance impairment with risk factors by using simple and multiple logistic stepwise regression.

Factors	Neck Pain with Static Balance Impairment	
	Crude OR (95% CI)	Adjusted OR (95% CI)
**Demographic**		
Gender		
Male	1.00	-
Female	2.12 (0.53 to 8.40)	
Age (years)		
<20	1.00	-
≥20	0.50 (0.53 to 8.40) *	
BMI (kg/m^2^)		
<20	1.00	-
≥20	0.49 (0.81 to 1.34) *	
Beck Depression Inventory or BDI (scores)		
<4	1.00	-
≥4	2.00 (0.71 to 5.65) *	
Dizziness Handicap Inventory or DHI (scores)		
<7	1.00	1.00
≥7	2.14 (0.76 to 6.04) *	2.02 (0.29 to 14.23)
**Smart phone** **and other visual display terminal** **use**		
Daily hours of smartphone use (hours)		
<4	1.00	1.00
≥4	20.8 (5.96 to 72.60) *	19.24 (4.72 to 78.48) **
Years of smartphone use (years)		
<4	1.00	1.00
≥4	5.83 (1.95 to 17.45) *	5.01 (1.12 to 22.38) **
Posture when using smartphone		
Sitting	1.00	-
Lying	1.9 (0.70 to 5.19)	
Daily hours of other visual display terminal use (hours)		
<2	1.00	1.00
≥2	2.57 (0.67 to 9.83) *	2.56 (0.34 to 19.12)
**Neck pain status**		
Neck pain or VAS (mm)		
mild pain (30–44 mm)	1.00	1.00
moderate pain (45–74 mm)	10.00 (1.25 to 79.95) *	7.66 (0.33 to 177.14)
Neck disability index or NDI (scores)		
<7	1.00	1.00
≥7	14.25 (3.04 to 66.86) *	12.91 (2.24 to 74.45) **
Pain regions (group)		
Right or left pain	1.00	1.00
Central pain	3.37 (1.20 to 9.45) *	1.80 (0.29 to 11.16)
Stage of neck pain (stage)		
Sub-acute	1.00	-
Chronic	2.64 (0.79 to 8.83) *	
Episode of neck pain (episode)		
<3	1.00	1.00
≥3	12.73 (3.37 to 48.00) *	0.61 (0.03 to 11.99)
**Cervical joint position sense error** (degrees)		
Right rotation		
<4.5	1.00	1.00
≥4.5	7.77 (1.66 to 36.37) *	5.65 (0.88 to 36.43)
Left rotation		
<4.5	1.00	-
≥4.5	6.33 (1.35 to 29.72) *	
**Craniocervical flexors test** (mmHG)		
<24	1.00	1.00
≥24	4.75 (1.00 to 22.44) *	3.70 (0.53 to 25.90)

Abbreviations: BMI, body mass index; kg/m^2^, kilograms per meter square; DHI, Dizziness Handicap Inventory; BDI, Beck Depression Inventory; VDT, visual display terminal; BESS, balance error scoring system, * *p*-value < 0.20, ** *p*-value < 0.0.

## Data Availability

The data will be available for anyone who wishes to access them for research purposes; contact should be made via the corresponding author: rungthiprt@gmail.com.

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
