# Peer review of "Prevalence of Static Balance Impairment and Associated Factors of University Student Smartphone Users with Subclinical Neck Pain: Cross-Sectional Study"

_ijerph, 2022, doi:10.3390/ijerph191710723_

Round 1

Reviewer 1 Report

The manuscript addresses an interesting topic to assess prevalence of static balance impairment in university students with subclinical neck pain and identify the associated risk factors. The study has several strengths, but there are a few issues that could make this manuscript better. I made a few comments and suggestions for each section of the manuscript, which is shown below.

Page

Section

Comments and Recommendations

1

Abstact

Regulating of smartphone use hours daily and restricting years of smartphone use may prevent static balance impairment and neck disability in smartphone users.

Ø  As a cross-sectional study, ‘may prevent’ is a little bit strong conclusion. What do you think?

2

Introduction

Previous studies in smartphone users have not yet determined the prevalence of static balance impairment in smartphone users.

Ø  Can you justify why static balance is meaningful to investigate over the previous dynamic balance in Introduction?

Ø  What is your rationale to explain the relationship between smartphone use and static balance?  It is important for the need for this study.

2

Methods

The inclusion criteria were: 1) subclinical or intermittent neck pain and mild to moderate neck dysfunction not taking treatment regularly, 2) 18-25 years of age, 3) BMI 30 kg/m2, 4) experienced in using smartphone > 6 months duration and daily smartphone screen time 2  hours [23], 5) a neck disability index (NDI) score range 5-14/50, 6) mild to moderate pain of visual analogue scale (VAS) 30 to 74 mm, and 7) voluntary participants who comprehend English.

Ø  Why should the participants have neck pain and smartphone experience to be included in this study? If someone don’t, he or she will have less static balance problem?

Ø  BMI 30 kg/m2 ? Does people with BMI > 30 kg/m2 have more static balance problem? You found less OR in BMI > 20 kg/m2.

4

Methods

For participants aged 18-25 years, the total BESS score of 15 is the cutoff point for balance impairment [25].

Ø  Where did you get this classification? Averaging the 20- to 39-year-old data and the healthy controls results in a BESS score of 10.93 errors in youth, who would often use the BESS based on [25]    Bell DR, Guskiewicz KM, Clark MA, Padua DA. Systematic review of the balance error scoring system. Sports Health 2011;3:287–95. https://doi.org/10.1177/1941738111403122.

4

Methods

The sub-tests of BESS were (i) double leg stance with feet together, (ii) single leg stance on the non-dominant foot and (iii) tandem stance with the non-dominant foot behind on firm and foam surfaces for each of three tests.

Ø  If possible, adding pictures for balance and other tests will help readers who are not so familiar with them.

Ø  Can you take an example how a person can get BESS score > 15?

Ø  How did you control the hip angle of the raised (dominant) side in the single leg standing, which is a major influencing factor?

6

Results

Participant characteristics are described in Table 1.

Ø  Can you report the mean, SD, range of participants characteristics?

Ø  How did you recruit so many age < 20, female, BMI < 20 participants than general college student population?

Ø  Why did you classify the BMI < and >= 20? Do you have a reference for this classification? Other cutoffs, e.g., BDI < 4, needs supporting references.

10

Discussion

No significant difference in static balance was found between male and female in the current study.

Ø  Is it fair to mention it, when you have very small number of male participants in your study? What do you think whether the result still be consistent when you have sex-balanced participants?

Ø  Please, add it as a limitation of your study.

10

Discussion

A BMI 30 kg/m2 is a confounding factor for balance control [63].

Ø  What do mean by a confounding factor? There are so many confounding factors for balance control. Is this why those peoples were excluded in this study?

Ø  Is it the right reference?

11

Effects on static balance control by years of smartphone use relate to visual fatigue which is one component of the balance control system.

Ø  This is only one possible scenario. Vision is important in balance control, but did your participants experience any visual fatigue? Furthermore, they were tested with their eyes closed.

Ø  Can you tell the fitness or activity level of your participants? That may explain better the high prevalence of the static balance impairment.

Author Response

Respond: Thank you for feedback. We have tried to respond your question and suggestion carefully.  The revised version of the manuscript is marked with green highlight changed.

Reviewer 2 Report

This study aims to identify the prevalence of static balance impairment and its associated factors. Some information is missing from the text and others need clarification:

1. Describe the justification for the association between neck pain and balance impairment;

2. Confidence intervals are too wide. Why it happened? 

3. Revise the study conclusion, as the cross-sectional design and findings do not support it ("regulating smartphone usage hours daily and restricting smartphone usage years may prevent neck and static balance impairment");

4. Please include the definition of subclinical, subacute, chronic, central and peripheral pain in the Introduction section;

5. Review the term "Beck Disability Index" on page 3 (line 100). Does it refer to the Beck Disability Inventory?

6. Inform how the sociodemographic and smartphone-related variables were obtained;

7. Describe all instruments used in the Methods sections (eg BDI, DHI);

8. Did the sample size calculation consider the multiple logistic regression analysis? Does the study have sufficient power for this analysis?

9. Include the reference number from Naing et al. (2006) cited on page 6 (line 239);

10. Why was the BMI dichotomized at 20?

11. Table 1: describe the label of the 3rd column, invert the percentage calculation to add 100% in columns 3 and 4;

12. Discussion (lines 308-310): Could you explain why you mentioned these deficits?

13. Discussion (lines 356-359): sitting and lying postures can be adopted when using the smartphone for the same subject. Please revise this argument;

14. Some non-significant findings were discussed. Please consider the results of the multivariate analysis.

Author Response

Respond: Thank you for comments. We have tried to respond your question and suggestion with great care. The revised version of the manuscript is almost marked with yellow highlight changed as follow:

Point 1: Describe the justification for the association between neck pain and balance impairment. The justification for the association between neck pain and balance impairment

Respond 1: Thank you for your question. From our review we can conclude as follow:

Smartphone screen time ≥ 2 hours per day lead to neck pain (Berolo et al., 2011). Balance control depends on proprioceptive information from mechanoreceptors, vestibular and visual input to the CNS (Clark et al., 2003; Hrysomallis, 2007; Salavati et al., 2007). The deficiency of cervical proprioceptive inhibition of nociceptors effect a loss of standing balance in chronic pain (McPartland et al., 1997). Moreover, long duration smartphone use increases visual fatigue (S. Park et al., 2017) and cervical joint position sense error (CJPSE) after smartphone use (Kim et al., 2013; Lee and Seo, 2014).

References

Berolo, S., Wells, R. P., & Amick, B. C. (2011). Musculoskeletal symptoms among mobile hand-held device users and their relationship to device use: A preliminary study in a Canadian university population. Applied Ergonomics, 42(2), 371-378. doi:https://doi.org/10.1016/j.apergo.2010.08.010

Clark, B. C., Manini, T. M., Thé, D. J., Doldo, N. A., & Ploutz-Snyder, L. L. (2003). Gender differences in skeletal muscle fatigability are related to contraction type and EMG spectral compression. Journal of Applied Physiology, 94(6), 2263-2272.

Hrysomallis, C. (2007). Relationship Between Balance Ability, Training and Sports Injury Risk. Sports Medicine, 37(6), 547-556. doi:10.2165/00007256-200737060-00007

Kim, Y.-G., Kang, M.-H., Kim, J.-W., Jang, J.-h., & Oh, J.-S. (2013). Influence of the duration of smartphone usage on flexion angles of the cervical and lumbar spine and on reposition error in the cervical spine. Physical Therapy Korea, 20(1), 10-17.

Lee, J., & Seo, K. (2014). The comparison of cervical repositioning errors according to smartphone addiction grades. Journal of physical therapy science, 26(4), 595-598.

McPartland, J. M., Brodeur, R. R., & Hallgren, R. C. (1997). Chronic neck pain, standing balance, and suboccipital muscle atrophy--a pilot study. Journal of manipulative and physiological therapeutics, 20(1), 24-29.

Park, S., Choi, D., Yi, J., Lee, S., Lee, J. E., Choi, B., . . . Kyung, G. (2017). Effects of display curvature, display zone, and task duration on legibility and visual fatigue during visual search task. Applied Ergonomics, 60, 183-193. doi:https://doi.org/10.1016/j.apergo.2016.11.012

Salavati, M., Moghadam, M., Ebrahimi, I., & Arab, A. M. (2007). Changes in postural stability with fatigue of lower extremity frontal and sagittal plane movers. Gait & Posture, 26(2), 214-218. doi:https://doi.org/10.1016/j.gaitpost.2006.09.001

Point 2: Confidence intervals are too wide. Why it happened? 

Respond 2: Thank you for indicating this point. Before conducted the study we have calculated the sample size according to us primary aim that was to evaluate the prevalence of static balance impairment of university student smartphone users with subclinical neck pain. So, this may affect to small sample size went the data were analyzed by using multiple logistic regression (Irala et al., 1997). According to this point we have added it as another list of limitation as follows:

“Third, although before conducting the study, we calculated the sample size according to prevalence as our main outcome, this samples may not be appropriate when the data were analyzed by multiple logistic regression for determining factors associated with balance impairment as the secondary outcome. Further research should consider se-lecting sample size by referencing a multiple logistic regression study to strengthen the results.”.

Reference

Irala, J. D., Fernandez-Crehuet Navajas, R., & Serrano del Castillo, A. (1997). Abnormally wide confidence intervals in logistic regression: interpretation of statistical program results. Revista Panamericana de Salud Pública2(4), 268-271.

Section: Limitation; Page: 11; Line: 440-445.

Point 3:  Revise the study conclusion, as the cross-sectional design and findings do not support it ("regulating smartphone usage hours daily and restricting smartphone usage years may prevent neck and static balance impairment");

Respond 2: Thank you so much for your suggestion. We have considered of your question, and we have rewritten this sentence as

“University student smartphone users with subclinical neck pain who were met at least one of risk factors should realize of their static balance impairment.”

Section: Abstract; Page: 1; Line: 26-27.

Point 4: Please include the definition of subclinical, subacute, chronic, central and peripheral pain in the Introduction section;

Respond 4: Thanks for your kind suggestion. We have added the detail your mention in our introduction as follows:

“Subclinical neck pain (SNP) refers to mild to moderate recurrent neck pain. Sub-clinical means that the participants have not yet taken treatment as the pain severity is nor severe. Stages of SNP may be within 7 days to 3 months (subacute SNP) or within more than 3 months (chronic SNP) [11-13]. It is classified according to the area identified by the subject who complained of pain on both sides of the neck, including central of the posterior part of the cervical spine from the superior nuchal line to the T1 spinous process, as central SNP. Right SNP is identified by the subject who complained of neck pain only on right side of the neck and left SNP only on the left side of the neck [12]”.

References

  1. Lee, H.-Y., Wang, J.-D., Yao, G., & Wang, S.-F. (2008). Association between cervicocephalic kinesthetic sensibility and frequency of subclinical neck pain. Manual therapy, 13(5), 419-425. doi:https://doi.org/10.1016/j.math.2007.04.001
  2. Lee, H., Nicholoson, L. L., Adams, R. D., & Bae, S.-S. (2005). Body Chart Pain Location and Side-Specific Physical Impairment in Subclinical Neck Pain. Journal of Manipulative and Physiological Therapeutics, 28(7), 479-486. doi:https://doi.org/10.1016/j.jmpt.2005.07.004
  3. Lee, H., Nicholson, L. L., & Adams, R. D. (2004). Cervical Range of Motion Associations With Subclinical Neck Pain. Spine, 29(1), 33-40. doi:10.1097/01.brs.0000103944.10408.ba

Section: Introduction; Page: 1; Line: 46,47; Page: 2; Line: 48-53.

Point 5: Review the term "Beck Disability Index" on page 3 (line 100). Does it refer to the Beck Disability Inventory?

Respond 5:  Thank you for your comment. The term Back Disability Index refers to Back Disability Inventory. We have rewritten the sentence as follows:

“2.5.7. Beck depression inventory

The BDI is a 21 item, self-report rating inventory that assesses characteristic attitudes and depression symptoms; each item corresponds to a major depressive symptom in the preceding 2 weeks [63].”

Section: Methods; Page: 7; Line: 273-276.

Point 6: Inform how the sociodemographic and smartphone-related variables were obtained;

Respond 6: We obtained the sociodemographic and smartphone-related variables from the literature review of the previous studies that showed impact on balance control are as follows:

Factors influenced on balance control

Some anthropometric (BMI, weight, foot size and width, height and leg dominance) and demographic measures (gender, age, occupation, levels and types of physical and recreational activities) can influence directly on both neck pain and balance control  (Ferrari & Sciance Russell, 1999). Age is an important factor in assessing postural balance but not important in young adults (Cavalheiro, Almeida, Pereira, & Andrade, 2009; Hue et al., 2007; Kejonen, Kauranen, & Vanharanta, 2003; Vieira, Oliveira, & Nadal, 2009). Hypertension can also negatively affect balance by damaging the large arteries and decreasing microcirculation in specific functional areas in elderly adults (Abate et al., 2009; Acar, Demırbüken, Algun, Malkoç, & Tekın, 2015). Moreover, Diabetes type II (Field et al., 2008) and types of prescribed medication(s) such as any sedative drug and alcohol within the past 48 hours (G. Y. Kim, Ahn, Jeon, & Lee, 2012; Røgind, Lykkegaard, Bliddal, & Danneskiold-Samsøe, 2003) influence on central processing mechanisms and postural sway outcomes (Chien & Sterling, 2010; Sjöström et al., 2003; Stokell, Yu, Williams, & Treleaven, 2011). Gender showed poor and conflicting correlation with balance control (Siu & Tai Wing, 2013). The neck pain may change cerebellar-motor cortex interaction (Baarbé et al., 2018) which alters both neck and limb sensorimotor function and motor control (Uthaikhup, Jull, Sungkarat, & Treleaven, 2012). Even mild to moderate neck dysfunction of subclinical neck pain can impact sensorimotor function (Haavik & Murphy, 2011; Taylor & Murphy, 2008). The strength measures were positively correlated with balance scores, clearly demonstrating the link between muscle strength and balance ability (King et al., 2016). Head relocation accuracy showed positive correlation with postural stability (Siu & Tai Wing, 2013).

References

Abate, M., Di Iorio, A., Pini, B., Battaglini, C., Di Nicola, I., Foschini, N., . . . Abate, G. (2009). Effects of hypertension on balance assessed by computerized posturography in the elderly. Archives of Gerontology and Geriatrics, 49(1), 113-117. doi:https://doi.org/10.1016/j.archger.2008.05.008

Acar, S., Demırbüken, İ., Algun, C., Malkoç, M., & Tekın, N. (2015). Is hypertension a risk factor for poor balance control in elderly adults? Journal of Physical Therapy Science, 27(3), 901-904. doi:10.1589/jpts.27.901

Baarbé, J. K., Yielder, P., Haavik, H., Holmes, M. W. R., & Murphy, B. A. (2018). Subclinical recurrent neck pain and its treatment impacts motor training-induced plasticity of the cerebellum and motor cortex. PloS one, 13(2), e0193413. Retrieved from http://europepmc.org/abstract/MED/29489878

Cavalheiro, G. L., Almeida, M. F. S., Pereira, A. A., & Andrade, A. O. (2009). Study of age-related changes in postural control during quiet standing through Linear Discriminant Analysis. BioMedical Engineering OnLine, 8(1), 35. doi:10.1186/1475-925X-8-35

Chien, A., & Sterling, M. (2010). Sensory hypoaesthesia is a feature of chronic whiplash but not chronic idiopathic neck pain. Manual therapy, 15(1), 48-53. doi:https://doi.org/10.1016/j.math.2009.05.012

Field, S., Treleaven, J., & Jull, G. (2008). Standing balance: A comparison between idiopathic and whiplash-induced neck pain. Manual therapy, 13(3), 183-191. doi:https://doi.org/10.1016/j.math.2006.12.005

Haavik, H., & Murphy, B. (2011). Subclinical Neck Pain and the Effects of Cervical Manipulation on Elbow Joint Position Sense. Journal of Manipulative and Physiological Therapeutics, 34(2), 88-97. doi:https://doi.org/10.1016/j.jmpt.2010.12.009

Hue, O., Simoneau, M., Marcotte, J., Berrigan, F., Doré, J., Marceau, P., . . . Teasdale, N. (2007). Body weight is a strong predictor of postural stability. Gait & Posture, 26(1), 32-38. doi:https://doi.org/10.1016/j.gaitpost.2006.07.005

Kejonen, P., Kauranen, K., & Vanharanta, H. (2003). The relationship between anthropometric factors and body-balancing movements in postural balance. Archives of Physical Medicine and Rehabilitation, 84(1), 17-22. doi:https://doi.org/10.1053/apmr.2003.50058

Kim, G. Y., Ahn, C. S., Jeon, H. W., & Lee, C. R. (2012). Effects of the Use of Smartphones on Pain and Muscle Fatigue in the Upper Extremity. Journal of Physical Therapy Science, 24(12), 1255-1258. doi:10.1589/jpts.24.1255

King, G. W., Abreu, E. L., Cheng, A.-L., Chertoff, K. K., Brotto, L., Kelly, P. J., & Brotto, M. (2016). A multimodal assessment of balance in elderly and young adults. Oncotarget, 7(12), 13297-13306. doi:10.18632/oncotarget.7758

Røgind, H., Lykkegaard, J. J., Bliddal, H., & Danneskiold-Samsøe, B. (2003). Postural sway in normal subjects aged 20–70 years. Clinical Physiology and Functional Imaging, 23(3), 171-176. doi:10.1046/j.1475-097X.2003.00492.x

Siu, E. H. K., & Tai Wing, T. C. (2013). The relationship between cervical range of motion, head-repositioning accuracy, and postural stability in healthy adults. International Journal of Therapy and Rehabilitation, 20(1), 9-17.

Sjölander, P., Johansson, H., & Djupsjöbacka, M. (2002). Spinal and supraspinal effects of activity in ligament afferents. Journal of Electromyography and Kinesiology, 12(3), 167-176. doi:https://doi.org/10.1016/S1050-6411(02)00017-2

Stokell, R., Yu, A., Williams, K., & Treleaven, J. (2011). Dynamic and functional balance tasks in subjects with persistent whiplash: A pilot trial. Manual therapy, 16(4), 394-398. doi:https://doi.org/10.1016/j.math.2011.01.012

Taylor, H. H., & Murphy, B. (2008). Altered Sensorimotor Integration With Cervical Spine Manipulation. Journal of Manipulative and Physiological Therapeutics, 31(2), 115-126. doi:https://doi.org/10.1016/j.jmpt.2007.12.011

Uthaikhup, S., Jull, G., Sungkarat, S., & Treleaven, J. (2012). The influence of neck pain on sensorimotor function in the elderly. Archives of Gerontology and Geriatrics, 55(3), 667-672. doi:https://doi.org/10.1016/j.archger.2012.01.013

Vieira, T. d. M. M., Oliveira, L. F. d., & Nadal, J. (2009). An overview of age-related changes in postural control during quiet standing tasks using classical and modern stabilometric descriptors. Journal of Electromyography and Kinesiology, 19(6), e513-e519. doi:https://doi.org/10.1016/j.jelekin.2008.10.007

Point 7:  Describe all instruments used in the Methods sections (eg BDI, DHI);

Respond 7: Thanks for the care suggestion. We have described all instruments used including BDI, DHI in the Methods sections as follows:

“2.4.6. Dizziness handicap inventory                                                     

            The DHI was developed to measure the self-perceived level of handicap associated with the symptom of dizziness [58]. The DHI has 25 items with 3 response levels, sub-grouped into three domains: functional, emotional, and physical. The DHI has good correlation with specific objective measures of balance [59-61]. The DHI may be useful in identifying subjects with benign paroxysmal positional vertigo [61]; a total score of 0 to 30 indicates a mild dizziness handicap, a total of score of 31 to 60 indicates a moderate dizziness handicap, and a total score of 61 to 100 indicates a severe dizziness handicap [61]. The DHI is reliable (r = 0.92 to 0.97) and is a valid, comprehensively, and clinically useful tool to measure self-perceived handicap associated with the symptom of dizziness from a variety of causes [62]. The participants were screened with DHI questionnaires to exclude vestibular disorder if their DHI score > 30/100. 

2.5.7. Beck depression inventory

            The BDI is a 21 item, self-report rating inventory that assesses characteristic attitudes and depression symptoms; each item corresponds to a major depressive symptom in the preceding 2 weeks [63]. Internal consistency for the BDI ranges from 0.73 to 0.92 with a mean of 0.86 [64,65]. The BDI demonstrates high internal consistency, with alpha coefficients of 0.86 and 0.81 for psychiatric populations and non-psychiatric populations, respectively [64]. The BDI lasts approximately 10 minutes to understand the questions comprehensively and sufficiently [65]. In the present study, the BDI was used as a screening device if a participant’s BDI score > 30/63 to exclude depression symptoms.”

Reference

  1. Jacobson, G. P., & Newman, C. W. (1990). The Development of the Dizziness Handicap Inventory. JAMA Otolaryngology–Head & Neck Surgery, 116(4), 424-427. doi:10.1001/archotol.1990.01870040046011
  2. Kaufman, L. J., Brangwynne, C. P., Kasza, K. E., Filippidi, E., Gordon, V. D., Deisboeck, T. S., & Weitz, D. A. (2005). Glioma Expansion in Collagen I Matrices: Analyzing Collagen Concentration-Dependent Growth and Motility Patterns. Biophysical Journal, 89(1), 635-650. doi:https://doi.org/10.1529/biophysj.105.061994
  3. Treleaven, J., Jull, G., & LowChoy, N. (2005). Smooth pursuit neck torsion test in whiplash-associated disorders: relation-ship to self-reports of neck pain and disability, dizziness and anxiety. Journal of Rehabilitation Medicine, 37(4), 219-223.
  4. Whitney, S. L., Wrisley, D. M., Brown, K. E., & Furman, J. M. (2004). Is Perception of Handicap Related to Functional Per-formance in Persons with Vestibular Dysfunction? Otology & Neurotology, 25(2), 139-143.
  5. Enloe, L. J., & Shields, R. K. (1997). Evaluation of Health-Related Quality of Life in Individuals With Vestibular Disease Using Disease-Specific and General Outcome Measures. Physical Therapy, 77(9), 890-903. doi:10.1093/ptj/77.9.890
  6. Beck, A. T., Ward, C. H., Mendelson, M., Mock, J., & Erbaugh, J. (1961). An inventory for measuring depression. Archives of general psychiatry, 4(6), 561-571.
  7. Beck, A. T., Steer, R. A., & Carbin, M. G. (1988). Psychometric properties of the Beck Depression Inventory: Twenty-five years of evaluation. Clinical psychology review, 8(1), 77-100.
  8. Groth-Marnat, G. (1999). Financial efficacy of clinical assessment: Rational guidelines and issues for future research. Jour-nal of Clinical Psychology, 55(7), 813-824. doi:10.1002/(sici)1097-4679(199907)55:7<813::aid-jclp4>3.0.co;2-b

Section: Methods; Page: 6; Line: 261-265; Page: 7; Line: 266-281.

Point 8: Did the sample size calculation consider the multiple logistic regression analysis? Does the study have sufficient power for this analysis?

Respond 8: Thank you so much for your question. According to our primary aim was to investigate the prevalence of static balance impairment in neck pain university student with smartphone user. The sample size calculation was more emphasize on prevalence as the detail in the section of sample size calculation follows:

“2.5. Sample Size

The sample size estimation was conducted after the pilot study. When P (prevalence) was greater than or equal to 0.6 and 95%CI (1.96), d (precision) could be set to 0.1 [66]. The sample size of the study was 81. It was considered acceptable described by Naing et al. (2006) [67] and Puntumetakul et al. (2022) [68].”

Reference

  1. Naing, L. (2006). Practical Issues in Calculating the Sample Size for Prevalence Studies. Archives of Orofacial Sciences, 1, 9-14.
  2. Puntumetakul, R., Chatprem, T., Saiklang, P., Phadungkit, S., Kamruecha, W., & Sae-Jung, S. (2022). Prevalence and Associ-ated Factors of Clinical Myelopathy Signs in Smartphone-Using University Students with Neck Pain. International Journal of En-vironmental Research and Public Health, 19(8), 4890.

Section: Methods; Page: 7; Line: 282-286.

The above mentioned of format of sample size calculation also in accorded to the study of Puntumetakul et al., (2022) in the title “Prevalence and Associated Factors of Clinical Myelopathy Signs in Smartphone-Using University Students with Neck Pain” (Puntumetakul et al., 2022)

Moreover, our 95%CI was seemly wide, this may be due to small sample size. So, we have added this issue as one of our limitations as follows:

“Third, although before conducting the study, we calculated the sample size according to prevalence as our main outcome, this samples may not be appropriate when the data were analyzed by multiple logistic regression for determining factors associated with balance impairment as the secondary outcome. Further research should consider selecting sample size by referencing a multiple logistic regression study to strengthen the results.”

 Section: Limitation; Page: 11; Line: 440-445.

Point 9: Include the reference number from Naing et al. (2006) cited on page 6 (line 239);

Respond 9: Thanks a lot.  We have added the reference number for Naing et al. (2006).

Reference

  1. Naing, L. (2006). Practical Issues in Calculating the Sample Size for Prevalence Studies. Archives of Orofacial Sciences, 1, 9-14.

Section: Methods; Page: 7; Line: 285-286.

Point 10: Why was the BMI dichotomized at 20?

Respond 10: Thanks for your question. People with BMI > 30 kg/m2 have more static balance problem as it influences negatively on both static and dynamic balance control (Carral, J. M. C et al., 2019). We recruited the participants having BMI£ 30 kg/m2 in the current study. According to the socioeconomic statues of Myanmar, nearly half of the participants in my study were having BMI less than 20 kg/m2. So, I dichotomized BMI at 20 based on mean of the participants for included it into the initial model of multivariate logistic regression.

Reference

  1. Carral, J. M. C., Ayán, C., Sturzinger, L., & Gonzalez, G. (2019). Relationships between body mass index and static and dynamic balance in active and inactive older adults. Journal of geriatric physical therapy, 42(4), E85-E90.

Section: Methods; Page: 3; Line: 105-106.

Point 11: Table 1: describe the label of the 3rd column, invert the percentage calculation to add 100% in columns 3 and 4;

Respond 11: Thanks for your kind suggestion. We have described the label of the 3rd column. I already described the percentage calculation to add 100% in columns 3 and 4.

Table 1. Characteristics of participants (n = 81).

Factors

Total

Participants,

n (%)

Mean (SD)

Min-Max

Participants without BESS impairment,

n (%)

Participants with BESS impairment,

n (%)

Demographic

Age (years)

< 20

≥ 20

45 (55.56)

36 (44.44)

19 (0.51)

21 (1.64)

18-19

20-25

9 (20)

12 (33)

36 (80)

24 (67)

Gender

  Male

  Female

10 (12.35)

71 (87.65)

-

-

4 (40)

17 (24)

6 (60)

54 (76)

BMI (kg/m2)

    < 20

    ≥ 20

49 (60.49)

32 (39.51)

18.17 (1.27)

22.45 (2.11)

15.9-19.95

20.3-28.38

10 (20)

11 (34)

39 (80)

21 (66)

BDI (score)

< 4

     ≥ 4

44 (54.32)

37 (45.68)

1.72 (1.93)

8.29 (8.38)

0-8

4-17

14 (32)

7 (19)

30 (68)

30 (81)

DHI (score)

< 7

≥ 7

43 (53.09)

38 (46.91)

0.67 (0.07)

0.82 (0.06)

0-6

8-28

14 (33)

7 (18)

29 (67)

31 (82)

Smart phone and other visual display terminal use

Daily hours of smartphone use

< 4

≥ 4

24 (29.63)

57 (70.37)

2.88 (0.27)

4.83 (1.70)

2.5-3.5

2.5-8.0

 16 (67)

5 (9)

8 (33)

52 (91)

Years of smartphone use

< 4

≥ 4

33 (40.74)

48 (59.26)

2.67 (0.48)

4.88 (0.90)

2-3

4-7

15 (45)

6 (13)

18 (55)

42 (88)

Posture when using smartphone use     

Sitting

Lying

33 (40.74)

48 (59.26)

-

-

11 (33)

10 (21)

22 (67)

38 (79)

Daily hours of other visual display terminal us

< 2

≥ 2

60 (74.07)

21 (25.93)

0.38 (0.48)

3.29 (1.82)

0-1

2-9

18 (30)

3 (14)

42 (70)

18 (86)

Neck pain status

Visual analogue scale (mm)

< 44

≥ 44

60 (74.07)

21 (25.93)

33.83 (4.37)

50.19 (4.75)

30-44

45-64

21 (35)

0 (0)

39 (65)

21 (100)

NDI (score)

< 7

≥ 7

43 (53.09)

38 (46.91)

5.30 (0.46)

9.29 (1.74)

5-6

7-14

19 (44)

2 (5)

24 (56)

36 (95)

Site of neck pain

Central pain

Right or left    pain

29 (35.80)

52 (64.20)

-

-

9 (17)

11 (52)

43 (83)

10 (48)

Stage of neck pain

Subacute     Chronic

54 (66.67)

27 (33.33)

-

-

17 (31)

4 (15)

37 (69)

23 (85)

Episode

< 3

≥ 3

14 (17.28)

67 (82.72)

-

-

10 (71)

11 (16)

4 (29)

56 (84)

Cervical joint position sense error (degrees)

Right rotation

< 4.5

≥ 4.5

52 (64.20)

29 (35.80)

3.05 (0.88)

6.29 (1.84)

1.33-4

3.33-11.3

20 (35)

1 (4)

37 (65)

23 (96)

Left rotation

< 4.5

≥ 4.5

55 (67.90)

26 (32.10)

3.03 (0.85)

6.09(1.80)

1.33-4

4.5-11.3

21 (40)

0 (0)

32 (60)

28 (100)

Craniocervical flexors test (mmHg)     

< 24

≥ 24

59 (72.84)

22 (27.16)

26.47 (2.39)

17.64 (3.06)

24-30

12-20

2 (9)

19 (32)

20 (91)

40 (68)

Abbreviations: SD, standard deviation; BMI, body mass index; DHI, Dizziness Handicap Inventory; BDI, Beck Depression Inventory

Section: Results; Page: 8.

Point 12: Discussion (lines 308-310): Could you explain why you mentioned these deficits?

Respond 12: Thanks for your question. According to your question point 14, we have reconsidered of this issue, and we have decided to delete all non-significant variable.

Point 13: Discussion (lines 356-359): sitting and lying postures can be adopted when using the smartphone for the same subject. Please revise this argument;

Respond 13: Thanks for your question. According to your question point 14, we have reconsidered of this issue, and we have decided to delete all non-significant variable.

Point 14: Some non-significant findings were discussed. Please consider the results of the multivariate analysis.

Respond 14: Thanks for your feedback. Some non-significant findings were discussed. These were possible risk factors of neck pain and balance control according to the literature review of previous studies. However, regarding of your concern we have reconsidered of this issue, and we have decided to delete all non-significant variable at adjusted odd ratio.

Submission Date

19th August 2022
